**Data Availability Statement:** All relevant data are in the paper and its supporting information files.

**Funding:** This study was funded by grants from the Natural Science Foundation of China (NO:

# Exploring the growth trait molecular markers in two sheep breeds based on Genome-wide association analysis

**Mirenisa Tuersuntuoheti**[1,2�], **Jihu Zhang**[1,2�], **Wen Zhou**[1,2], **Cheng-long Zhang**[1,2], **Chunjie Liu**[1,2], **Qianqian Chang**[1,2], **Shudong Liu**[1,2]*

**1** College of Animal Science and Technology, Tarim University, Alar, China, **2** Tarim Science and Technology Key Laboratory of Xinjiang Production and Construction Corps, Alar, China

y These authors contributed equally to this work.
* liushudong63@126.com

## Abstract

Growth traits are quantitative traits controlled by multiple micro-effect genes. we identified molecular markers related to sheep growth traits, which formed the basis of molecular breeding. In this study, we randomly selected 100 Qira Black sheep and 84 German Merino sheep for the blood collection the jugular vein to genotype by using the Illumina Ovine SNP 50K Bead Chip. quality control criteria for statistical analysis were: rejection detection rate < 90% and minimum allele frequency (MAF) < 5%. Then, we performed Genome-wide association studies (GWAS) on sheep body weight, body height, body length, and chest circumference using mixed linear models. After getting 55 SNPs with significant correlation, they were annotated by reference genome of *Ovis aries* genome (Oar_v4.0) and We obtained a total of 84 candidate genes associated with production traits (*BMPR1B*, *HSD17B3*, *TMEM63C*, etc.). We selected *BMPR1B* for population validation and found a correlation between the *FecB* locus and body weight traits. Therefore, this study not only supplements the existing knowledge of molecular markers of sheep growth traits, but also has important theoretical significance and reference value for the mining of functional genes of sheep growth traits.

## Introduction

Growth traits have economical importance in sheep production. To increase the income of herdsmen and speed up sheep breeding, molecular breeding technology has been used in sheep breeding. With the publication of the HapMap Project, Genome Sequencing Project, and the emergence of commercial high-density SNP chips (cattle 54 K, pigs 60 K, chickens 60 K, horses 60 K, and sheep 50 K) and genotyping techniques, GWAS has become an important method to detect the candidate genes for complex quantitative traits in livestock and poultry.

GWAS studies on different traits were carried out in sheep, including the angular type [1], polyangular traits [2], coat color [3], and polyparous traits [4]. Zhang *et al.* [5] used the ovine 50 K SNP chip to conduct GWAS on 329 pure sheep (Sunite sheep, German Merino sheep, and Dolper sheep). Candidate genes for growth and meat traits (birth weight, weaning weight, weight at 6 months of age, eye muscle area, backfat thickness, daily gain before weaning, daily

32060743), Bintuan Science and Technology Program (NO: 2022CB001-09), and Autonomous Region Agricultural Area High-efficiency Mutton Sheep Breeding and Promotion Technology System Project(NO: xjnqry-g-2023). The funders had no role in study design, data collection and analysis, decision to publish, or preparation of the manuscript.

**Competing interests:** The authors have declared that there are no competing interests.

gain after weaning, daily gain, body height, chest circumference, and tube circumference) were identified. Then, 36 SNPs were found to be significantly correlated with seven traits (weaning weight, 6-month-old weight, daily gain before weaning, daily gain after weaning, daily gain, chest circumference, and tube circumference). Among them, 10 SNPs were significantly correlated with the daily gain after weaning, and five candidate genes (*MEF2B*, *RFXANK*, *CAMKMT*, *TRHDE*, and *RIPK2)* were found. These genes have also been further verified in other variety and by selective signal detection methods [6–8]. Gholizadeh *et al.* [9] conducted GWAS analysis of the birth weight, weaning weight, six-month-old weight, and one-year-old weight of 96 Baluchi sheep, and found 13 SNPs with significant presence at the chromosome level. Through gene annotation, the *STRBP* and *TRAMIL1* genes found to be birth weight-related. Furthermore, *APIP* and *DAAM1* were associated with weaning weight, *PHF15*, *PRSS12* and *MAN1A1* were associated with 6-month weight, while *SYNE1*, *WAPAL*, and *DAAM1*were associated with the 1-year weight. In 2015, Matika *et al.* [10] used the X-ray computed tomography (CT) to conduct GWAS analysis on the carcass traits of 600 Scottish Blackface sheep. A series of candidate genes affecting muscle, bone, and fat traits in Scottish Blackface sheep were identified on chromosomes 1, 3, 6 and 24. The genes on chromosome 3 were *EFEMP1*, *SPTBN1*, and *FSHR*, while those on chromosome 6 were *RXFP2*, *ABCG2*, *NCAPG*, *OST/SPP1*, *MEPE*, *IBSP* and *LCORL*. The genes on chromosome 24 were *SH2B1*, *MAPK3*, *TBX6*, *KIF22*, *IL4R*, and *IL21R*. Tao *et al.* [4] found that *RXFP2* was candidate gene affecting the trait of Zeller black sheep, while the candidate genes that screened in GWAS and affected the body shape trait were: *FAM19A2*, *DISP1*, *KIF1C*, *PDGFD*, *S1PR3*, *RUNX1T1*, *S1PR1*, *TBC1D4*, *CXCL12*, *MFAP1*, *DIAPH3*, *ZSCAN4*, and *ADGRB3*.

Although GWAS related to sheep growth traits have been previously reported, growth traits are quantitative traits that are affected by multiple genes. In order to search for related molecular markers, we need to supplement the currently available sheep growth and development related QTLs database. In this study, we used the Illumina 50K middensity chip to conduct the GWAS analysis on body weight, body height, body length, and chest circumference of Qira Black sheep and German Merino sheep, to explore the genetic markers and functional genes affecting the sheep growth traits.

## Materials and methods

### Experimental animals

All the information required for this study was provided by the Animal Ethics Committee of the College of Animal Science and Technology of Tarim University, Xinjiang, China. The phenotype dataset comprised 273 body weight, body height, body length, and chest circumference records from 1,200 ewes that were collected from 2020 to 2022.

A total of 84 German Merino sheep (from Kezhou Breeding Farm) and 100 Qira Black sheep (from Qira County Tianjin Aoqun Livestock Breeding Sheep Farm) among 1,200 ewes were genotyped using the Illumina 50K SNP panel for 54,241 markers. The SNP herd verification included 89 Qira Black sheep from Qira County Tianjin Aoqun Livestock Breeding Sheep Farm. All experimental animals were randomly selected adult healthy sheep.

The body height, body length, weight, and chest circumference of the experimental animals were measured using electronic scales, measuring rods, and tape measures.

### Genomic DNA extraction and quality control

Blood was collected from the evaluated sheep by puncturing the jugular. We used tubes containing $K_2$-EDTA as an anticoagulant. The samples were stored at 4°C for DNA was extracted by the phenol-chloroform method [11]. After extraction, the quality of the DNA samples was

assessed by determining the A260/280 in the biophotometer (type: DS-11,DeNovix, USA). The samples were accepted only when the A260/280 values were between 1.8 and 2.0. Then, the DNA was quantified and samples were diluted to a minimum and a maximum concentration of 50 ng/μL and 150 ng/μL, respectively, for the Illumina Ovines SNP 50 chip.

The BeadScan software converts the images and the Genome Studio software processes the data. The Plink 1.07 [12] software was used for data quality control. The quality control criteria were: the individual samples with a detection rate of < 95% were removed, the SNPs detection rate was < 90%, the minimum allele frequency was < 5%, and the $p$-value of Hardy-Weinberg equilibrium test was < $1 \times 10^{-6}$.

## Genetic relationship matrix and principal component analysis

GCTA software [13] was used for principal components analysis (PCA) analysis of filtered SNPs data, which was based on the genetic correlation identification between individuals and it also represented the main components of population structure. The genetic matrix was calculated by the KING (http://people.virginia.edu/~wc9c/KING/manual.html) software and mapped by R language.

## Statistical methods and models

GWAS analysis was performed using the TASSEL 5.0 software (https://www.maizegenetics.net), and was based on a mixed linear model (MLM). GWAS analysis of four body weight traits in 184 sheep was performed using genotypes obtained from Illumina OvineSNP50 microarray, with full consideration of population stratification and individual relationship. The statistical model is as follows:

$$Y = \mu + xg + wb + mu + e$$

where $y$ is the vector f the observed population phenotype; $\mu$ is the mean vector; $g$ is the fixed environmental effect vector; $b$ is the SNP effect vector; $u$ is the multigene effect vector; $x$, $w$, and $m$ are the association matrices of $g$, $b$, and $u$, respectively; and $e$ is the residual.

It was assumed that both $u$ and $e$ obeyed normal distribution. $e$ is the random vector of P ´1, the residual in $a$ general sense, $e \sim$ N (0, $R$), $R = G\,s^2$, $s^2$ is the residual variance, and $G$ is the residual correlation matrix. $u$ is the random vector of M ´1, called the random effect of $u$, $u \sim$ N (0, $Y$), $Y = s^2\,K$, $s^2$ is the additive genetic variance of Genome-wide association analysis or multi-locus analysis; $K$ is the correlation matrix of kinship. Threshold is determined by the Bonferroni correction method. SNPs are considered to be strictly Genome-wide significant at $P < 0.05/$ N and potentially significant at $P < 0.01/$ N. N is the number of bits after quality control of the genome chip.

## Candidate gene annotation

Statistically significant SNPs annotations referencing the genome of *Ovis aries* genome (Oar_v4.0) were obtained. The functional annotation of candidate genes was performed referencing the NCBI databases (http://www.ncbi.nlm.nih.gov/gene) and OMIM database (http://www.ncbi.nlm.nih.gov/omim).

## Primer design and gene sequencing

Primers were designed using the Primer Premier 6.0 (http://www.premierbiosoft.com) software according to the sequence of sheep *BMPRIB* available in the GenBank database (*BMPRIB* GenBank Landing Number: NM_001009431). The Primer Premier 6.0 software was used to

**Table 1. Primer information of *BMPRIB* Gene.**

| Gene | primer sequence (5'-3') | fragment length/bp | annealing temperature/˚C |
|---|---|---|---|
| *BMPR1B* | F: TGGAAAAGGTCGCTATGGGG | 192 | 60 |
| | R: AGCTAGGAAACCCTGAACATCG | | |

design primers for the obtained full-length sequence of the *BMPR1B*. The primer information is shown in Table 1. All primers were synthesized by the Shanghai (China) Sheng gong Bioengineering Company.

The 25 μL PCR reaction system contained: 2× Taq Master Mix—12.5 μL, upstream and downstream primers—1 μL (0.5 μmol/L), DNA template 1 μL (50 ng/μL), ddH$_2$O - 9.5 μL. The PCR amplification reaction conditions were: pre-denaturation at 94˚C for 5 min; denaturation at 94˚C for 45 s, annealing temperature for 30 s, extension at 72˚C for 45 s, a total of 35 cycles; the final extension at 72˚C for 5 min; followed by storage at 4˚C. PCR products were detected by electrophoresis and sent to the Shanghai Shenggong Company for sequencing.

## Statistical analysis of data

Sequencing results were compared on DNASTAR [14] to search for polymorphic loci. Using the SPSS 26.0 [15] software, a one-way ANOVA combined with Duncan's model was used to analyze the effects of different loci and genotypes on growth traits of Qira Black sheep ewes. The results were expressed as mean ± standard deviation and $p < 0.05$ was used as the criterion for significance of difference.

# Results and analysis

## Quality control and data analysis

After quality control, we obtained 46,871 SNPs. We conducted a descriptive analysis on the observed values, thus providing a preliminary analysis of their inherent information to describe the overall characteristics of groups. Descriptive statistical analysis is the basic requirement and premise of the follow-up statistical analysis. The growth traits of the resource population in this study included body weight, body height, body length, and chest circumference. Table 2 shows the analysis indexes of growth traits of population resource sheep.

## Sample population structure analysis

Fig 1 shows the heat map of the genetic relationship matrix of 100 Qira Black sheep and 84 German Merino sheep. The heat map of the kinship matrix indicates the distance of the kinship between each sheep using different shades of color. The lighter the color, the smaller the value, the more distant is the relationship. Therefore, as seen from Fig 1, two dark areas represent two varieties.

**Table 2. Descriptive statistics of population growth traits of Qira Black sheep and German Merino sheep.**

| Traits | sample size | Mean | standard deviation | Maximum value | minimum value |
|---|---|---|---|---|---|
| Body height/kg | 184 | 102.97 | 8.88 | 70.00 | 27.00 |
| Body weight/cm | 184 | 70.63 | 6.00 | 92.00 | 51.00 |
| Body length/cm | 184 | 76.16 | 15.24 | 103.00 | 50.00 |
| Chest circumference/cm | 184 | 47.04 | 15.60 | 135.00 | 76.00 |

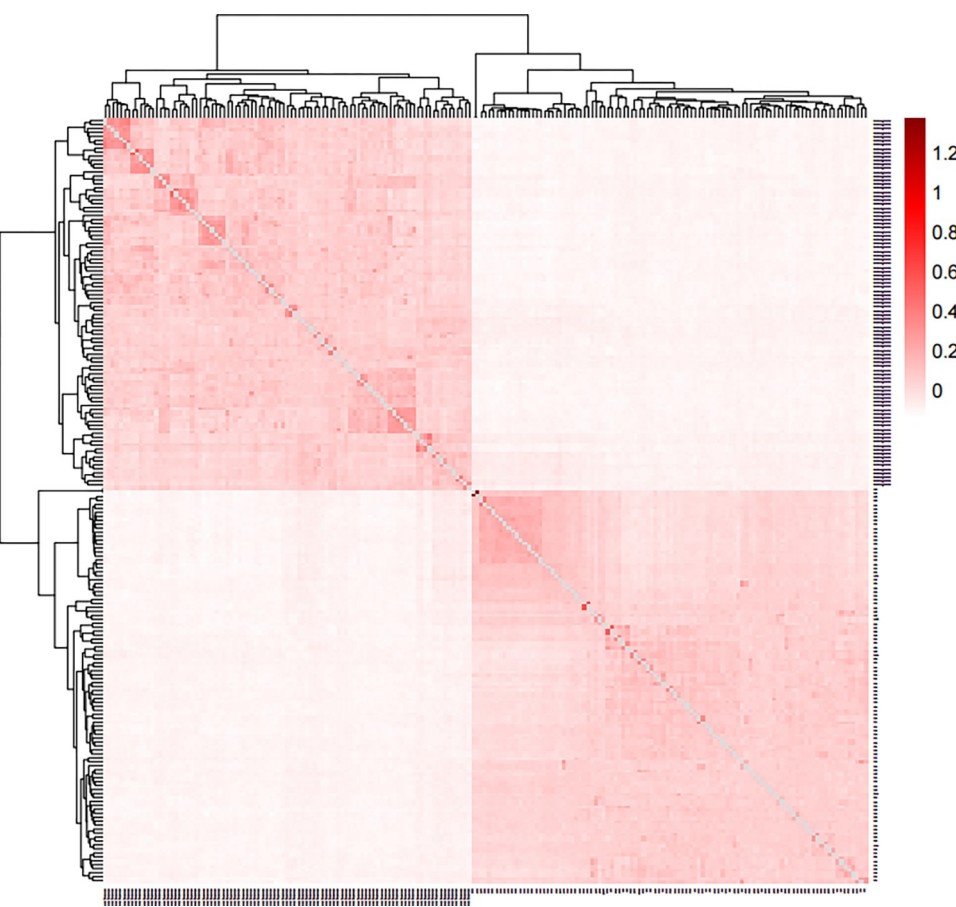

**Fig 1. Heat map of genetic relationship matrix between Qira Black sheep and German Merino sheep.**

Fig 2 shows the genetic structure distribution in the two sheep breeds, which causes the difference in breeds structure post selection. Therefore, population stratification should be considered and corrected in subsequent Genome-wide association analysis.

## Genome-wide association analysis and gene annotation results

GWAS analysis results showed that 55 SNPs were significantly correlated with the four growth traits (body weight, body height, body length and chest circumference) at the chromosomal level. Using genome of *Ovis aries* genome (Oar_v4.0) and public database information like the NCBI, we compared and annotated the SNPs loci with significant associations with productive traits obtained in the GWAS analysis results for different growth traits (S1–S4 Tables).

The results were analyzed by GWAS, and the original data comprised three body size traits and one body weight trait. Fig 3 shows the Manhattan diagram of four growth traits, with 46,871 SNPs being distributed on 27 chromosomes for association analysis. The significance level threshold after correction was $1.0 \times 10^{-6}$.

There were 55 SNPs significantly associated with these traits, which were located on 19 chromosomes. SNPs were found to be significantly associated with the body weight, body height, body length, and chest circumference, and were found on chromosomes 1, 2, 3, 5, 6, 7, 8, 9, 12, 13, 17, 18, 19, 21, 22, 24, and 26.

From the Table 3, we can see that significant SNPs were found through GWAS analysis and corresponding basic information. We screened the candidate genes for related traits in body

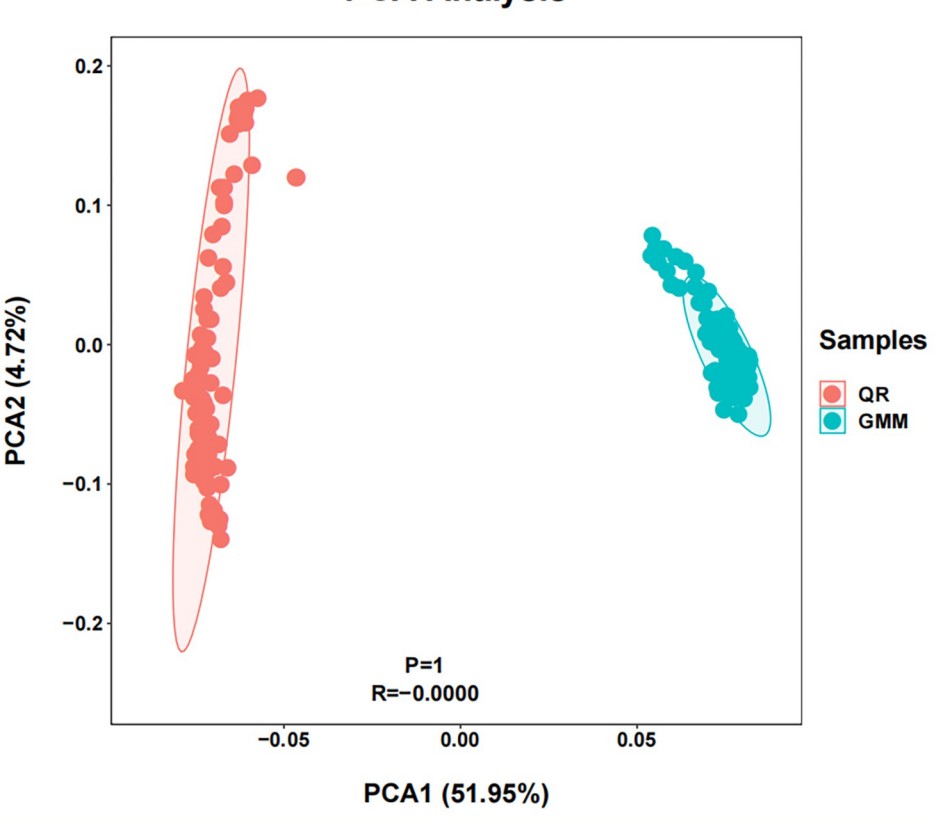

**Fig 2. PCA group results of Qira Black sheep and German Merino sheep.** Note: The red part (QR) is Qira Black sheep; The green part (GM) is German Merino sheep.

length, body height, body weight, and chest circumference. According to the *P*-value of $1.0 \times 10^{-6}$ locus, we found 55 SNPs related to these traits along with 84 nearby genes (Tab 3). The names, physical locations, and neighboring genes of SNPs significantly related to growth traits are shown in Table 3.

### Population stratification

Fig 4 shows the Q-Q Plot of four phenotypic traits. The observed and expected GWAS values of the sheep body weight, body height, body length, and chest circumference are shown in Fig 4. The dotted line represents the SNP distribution under the original hypothesis to indicate that the SNP is not related to the research characteristics. The strong deviation between the observed values and expected *p*-values of the four Q-Q plots indicated that there were more SNPs significantly associated with the traits than expected.

### Population validation results

### Amplification results of PCR products

PCR amplification results showed that the amplified region of *BMPRIB* gene was 192 bp (Fig 5), which was consistent with the expected target band size, and it also lacked heterozygous bands, which meets the requirements of subsequent tests.

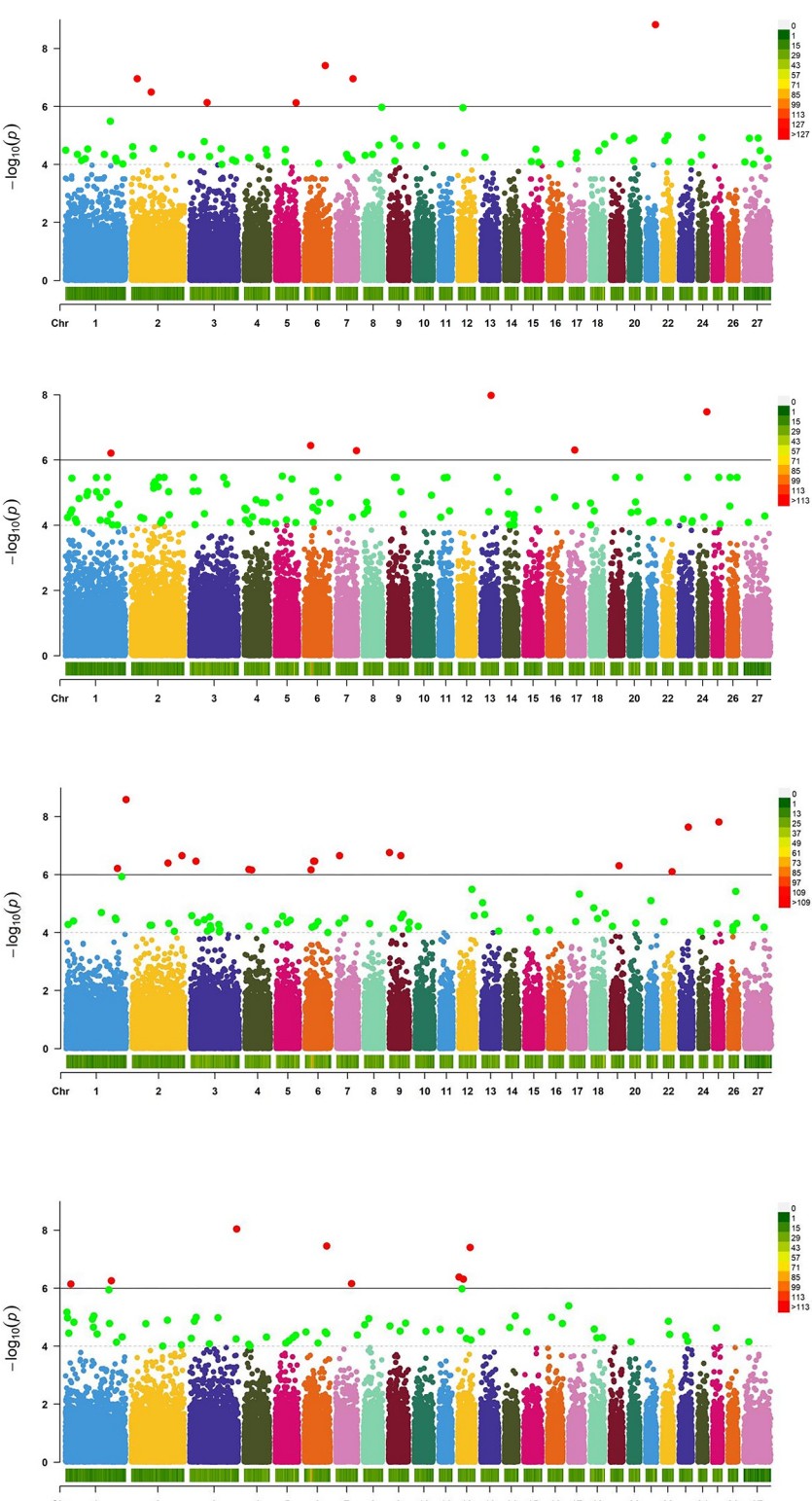

**Fig 3. Manhattan plot of Genome-wide association analysis of growth traits between Qira Black sheep and German Merino sheep.** Note: A for Body length; B for Body height; C for Body weight; D for the chest circumference. The horizontal red line in the figure represents the Genome-wide significance level threshold, and SNPs located above the red line indicate reaching the Genome-wide significance level.

**Table 3. Genome-wide association analysis results and candidate genes of growth traits of Qira Black sheep and German Merino sheep.**

| Traits | SNP site | Chromosome | position | *P*- Value | Candidate gene |
|---|---|---|---|---|---|
| Body length | OAR2_30011328 | 2 | 30011328 | 1.10E-07 | HSD17B3, RAD26L |
| | s22353.1 | 7 | 85421164 | 1.11E-07 | TMEM63C |
| | OAR12_24704107 | 12 | 24704107 | 1.10E-06 | TLR5 |
| | OAR1_223367286.1 | 1 | 223367286 | 3.25E-06 | PDHA1, OTOL1 |
| | s47244.1 | 21 | 48748297 | 1.50E-09 | LSP1 |
| | OAR6_104486245.1 | 6 | 104486245 | 3.88E-08 | TMEM128, OTOP1 |
| | OAR2_98995823.1 | 2 | 98995823 | 3.21E-07 | SMS, GLIPR2 |
| | s35034.1 | 3 | 85779808 | 7.29E-07 | ATL2, SET |
| | OAR5_103269743.1 | 5 | 103269743 | 7.45E-07 | EFNA5 |
| | s56894.1 | 8 | 91150511 | 1.07E-06 | PDCD2 |
| Body height | OAR2_117457778 | 2 | 117457778 | 4.63E-06 | PRSS58, NAB1 |
| | OAR2_111010640 | 2 | 111010640 | 7.40E-06 | PALLD |
| | OAR3_44825376.1 | 3 | 41906687 | 9.03E-06 | MEIS1, GSR |
| | OAR13_50641702.1 | 13 | 50641702 | 1.05E-08 | PANK2, MAVS |
| | s08282.1 | 24 | 43414683 | 3.32E-08 | ZFAND2A |
| | OAR6_32186608.1 | 6 | 32186608 | 3.62E-07 | CFDP2 |
| | s52728.1 | 17 | 29611538 | 4.97E-07 | HSPA4L |
| | OAR7_102639800.1 | 7 | 102639800 | 5.18E-07 | TRIM69 |
| | s06270.1 | 1 | 226537720 | 6.18E-07 | IQCJ,MFSD1 |
| Body weight | OAR22_44647841 | 22 | 44647841 | 7.98E-07 | ADAM12 |
| | OAR1_276613221.1 | 1 | 276613221 | 1.18E-06 | PRDM9 |
| | OAR12_65833655 | 12 | 65833655 | 3.25E-06 | PDC, PTGS2 |
| | s45909.1 | 26 | 34804161 | 3.83E-06 | SFRP1, GOLGA7 |
| | OAR17_49471964 | 17 | 49471964 | 4.73E-06 | TMEM132C, RPS14, RPL10A |
| | OAR21_22389818.1 | 21 | 22389818 | 7.89E-06 | ANO5, NELL1 |
| | OAR13_5856310.1 | 13 | 5087054 | 9.40E-06 | BTBD3 |
| | s45224.1 | 1 | 298080822 | 2.59E-09 | PRDM9 |
| | s08643.1 | 25 | 27371527 | 1.54E-08 | SLC29A3, UNC5B, CDH23 |
| | OAR23_42160314.1 | 23 | 42160314 | 2.29E-08 | PPP4R1, RAB31 |
| | OAR9_989389.1 | 9 | 989389 | 1.74E-07 | KCNQ5, CFDP2 |
| | s70589.1 | 2 | 248197546 | 2.22E-07 | PADI1 |
| | s72029.1 | 7 | 15078481 | 2.22E-07 | ITGA11, UBR2 |
| | OAR9_57398243.1 | 9 | 57398243 | 2.22E-07 | FABP5, PMP2 |
| | s42459.1 | 3 | 27290601 | 3.44E-07 | OSR1, TTC32 |
| | OAR6_42528741.1 | 6 | 42528741 | 3.44E-07 | GBA3, PPARGC1A |
| | OAR6_48236433.1 | 6 | 48236433 | 3.44E-07 | MAGEA13P, NAP1L1 |
| | OAR2_179242768.1 | 2 | 179242768 | 4.00E-07 | POMK, DPP10 |
| | OAR19_40476036.1 | 19 | 40476036 | 4.92E-07 | RPL17 |
| | s33133.1 | 1 | 255168152 | 6.11E-07 | ACAD11, DNAJC13 |
| | OAR4_20451557.1 | 4 | 20451557 | 6.63E-07 | TMEM106B, VWDE |
| | s17623.1 | 6 | 29513430 | 6.83E-07 | BMPR1B |
| | OAR4_33022606_X.1 | 4 | 33022607 | 6.97E-07 | CROT |

(*Continued*)

**Table 3.** (Continued)

| Traits | SNP site | Chromosome | position | P- Value | Candidate gene |
|---|---|---|---|---|---|
| Chest circumference | OAR12_5223258 | 12 | 5223258 | 4.12E-07 | DNAJC13 |
| | OAR1_215887929.1 | 1 | 215887929 | 1.14E-06 | CCDC115 |
| | OAR17_506915.1 | 17 | 506915 | 4.06E-06 | CPE |
| | OAR1_7026065 | 1 | 7026065 | 6.69E-06 | MROH2A, HEATR7B1 |
| | OAR1_141298457 | 1 | 141298457 | 8.99E-06 | ABCC1 |
| | OAR3_233748114.1 | 3 | 233748114 | 9.15E-09 | RABL2A |
| | OAR6_111740006.1 | 6 | 111740006 | 3.50E-08 | LDB2 |
| | OAR12_60622022.1 | 12 | 60622022 | 3.92E-08 | IER5, CACNA1E |
| | OAR12_26860344.1 | 12 | 26860344 | 4.84E-07 | ENAH |
| | OAR1_228809955.1 | 1 | 228809955 | 5.58E-07 | LEKR1 |
| | OAR7_78796628.1 | 7 | 78796628 | 6.96E-07 | SLC8A3 |
| | OAR1_27269648.1 | 1 | 27269648 | 7.23E-07 | SCP2 |
| | s74379.1 | 12 | 19596293 | 1.04E-06 | SPATA17, HSPA8 |

## Sequencing verification and mutation site identification

Sequencing results showed that *BMPR1B* gene had a mutation site (Fig 6): g.431965A > G; at the 431965A > G mutation, resulting in three genotypes: AA, AG, and GG.

## Association analysis of *BMPRIB* Gene polymorphism and growth traits of Qira Black sheep

As can be seen from Table 4, the *BMPR1B* gene g.431965A>G locus in the Qira Black sheep, The weight of the AA genotype at the G locus was significantly higher than those of the AG and GG genes ($p < 0.05$). The weight of the AG genotype did not differ significantly from that of the GG genotype ($P > 0.05$).

## Discussion

The false positive phenomenon of GWAS analysis was mainly due to population stratification, which eventually gave rise to multiple SNPs loci associated with the concerned traits in the GWAS analysis results [16]. To reduce these false-positive results, population stratification and relatedness among individuals were fully considered. We used PCA and breed effect to solve the population stratification phenomenon. As can be seen from the QQ-plot, there was no population stratification phenomenon in the corrected population, and the GWAS analysis results based on the MLM were relatively reliable. Additionally, the TASSEL 5.0 software provides a kinship matrix, two correction a group the genetic relationship between the two individuals, to improve the effectiveness of the GWAS analysis. In this study, we performed a GWAS on the Qira Black sheep and German Merino sheep via genotyping data by using a medium-density chip containing 46,871 SNPs. We detected 55 SNPs that were significantly associated with growth traits. From the genomic screening information, we found over 10% of the genes to be located in the sheep chromosomes 1, 2, 6, and 12. Among them, 10 loci were located on chromosome 1 (OAR1_223367286.1, S06270.1, S33133.1, etc.). There were 6 SNPs loci on chromosome 2, 6, and 12 (OAR2_179242768.1, OAR6_111740006.1, OAR12_60622021.1, etc.) We obtained a total of 84 genes through gene annotation. Based on gene enrichment analysis, we can preliminarily infer that these loci were important molecular markers that affected the growth and meat production traits of sheep.

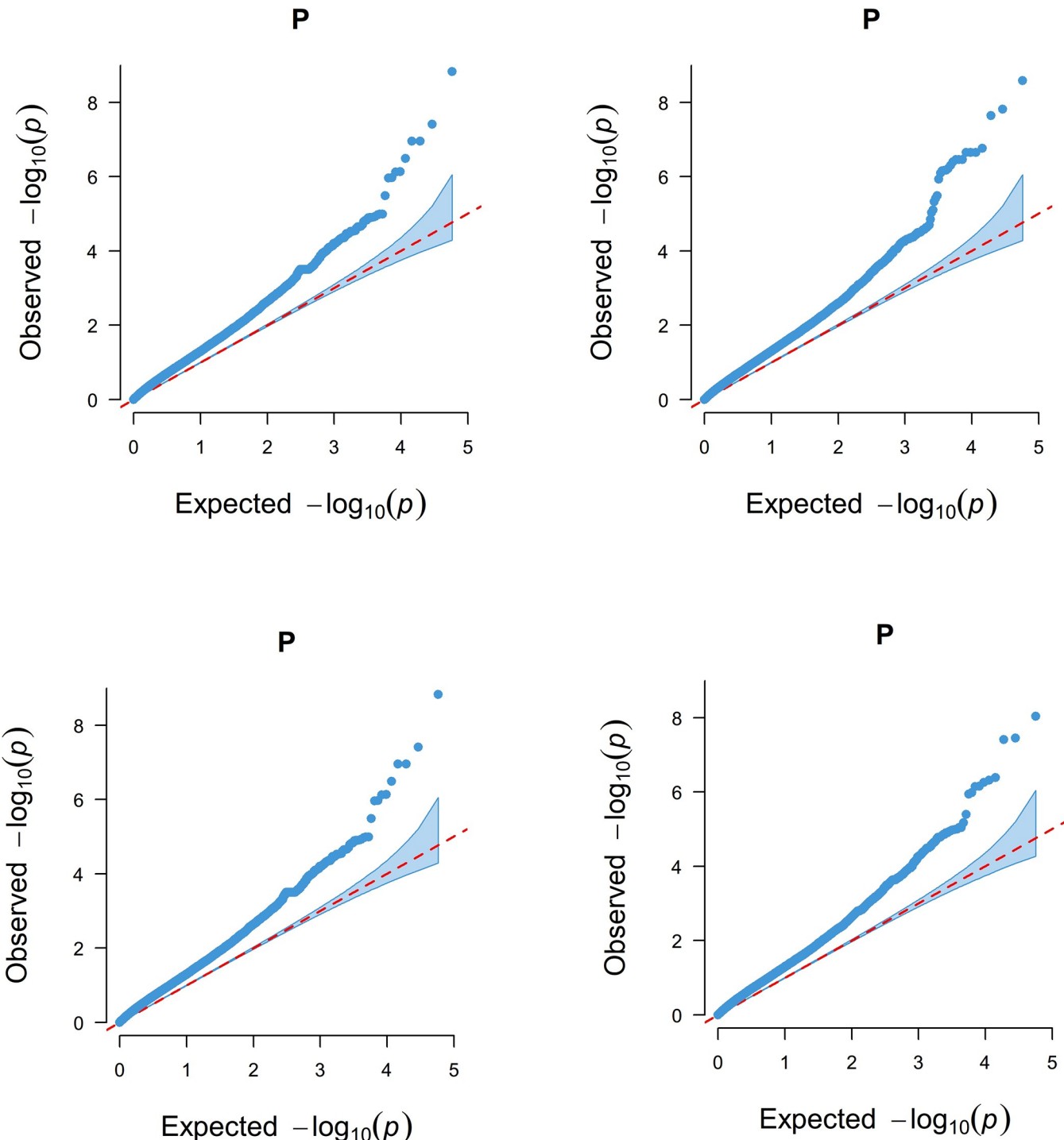

**Fig 4. Quantile-quantile (Q-Q) plots of Genome-wide association results for 4 traits.** Note: A for Body length; B for Body height; C for Body weight; D for the chest circumference.

Zhang *et al.* [5] had obtained 36 significant SNPs for seven traits, while we obtained 55 significant SNPs for 4 traits (body weight, body height, body length, and chest circumference), which enriched the SNPs related to sheep growth and development. Furthermore, Jiang *et al.*

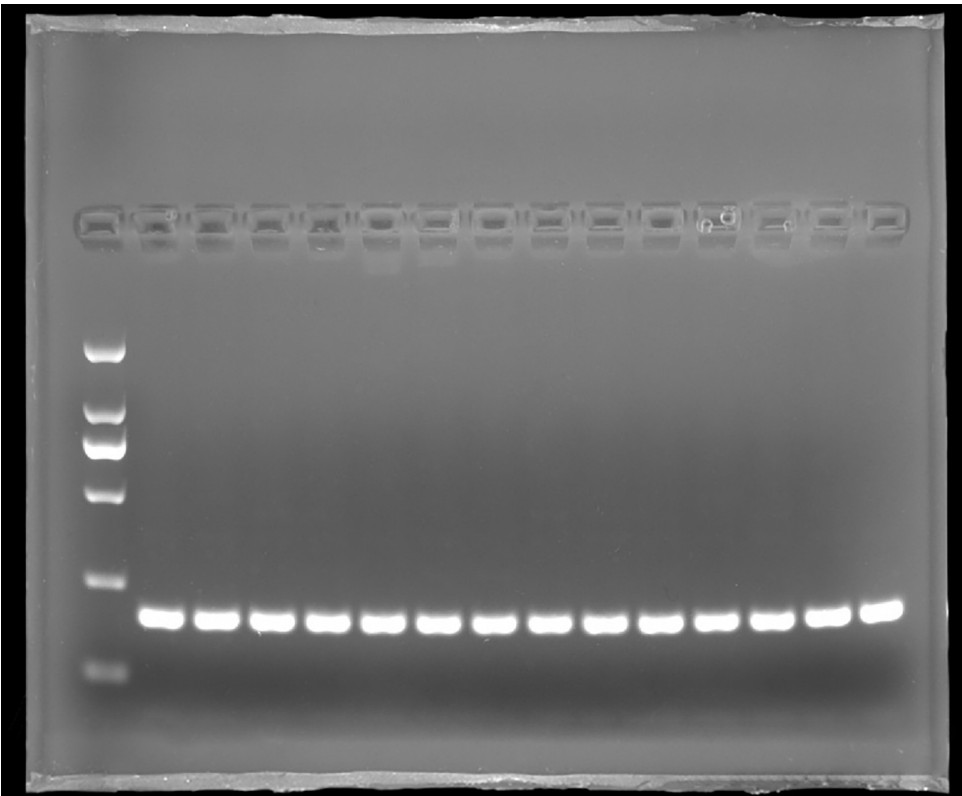

**Fig 5. PCR amplification of *BMPRIB* Gene.** M: DNA standard DL 2000; 1–14: Amplification product of the *BMPRIB* gene.

[17] obtained 5 SNPs related to body height and 4 SNPS related to chest circumference in Hu sheep, whereas Hawlader *et al*. [18] obtained 39 SNPS related to body height in Australian Merino sheep. However, we identified 9 and 13 SNPs correlated with body height and chest circumference in this study, respectively. Since we had selected different germplasm resources for this study, we obtained the results at different loci. Although some of the annotated genes have already been previously reported, we found genes related to growth and development traits, including *POMK*, *MEIS1*, *CROT*, *BMPR1B*, *FABP5*, *SPATA17*, etc. Based on the NCBI and Gene Cards database annotations, we found that among the meat quality traits, genes related to growth traits mainly included *POMK*, *BMPR1B*, *FABP5*, *RPL17*, *EFNA5*, *NELL1* and *PPARGC1A*.

The *protein O-mannose kinase* (*POMK*), located on chromosome 26 of sheep (37073963~37090331 bp), encodes a vital glycosylated and functional dystrophic glycan complex, which is critical to the extracellular matrix composition. Zhu *et al*. [19] reported that the *POMK* lesions can result in an abnormal functioning α-dystroglycan, which affects the development of the muscle and brain, ultimately resulting in congenital muscular dystrophy. The underlying mechanism of this phenomenon is still unclear. Since the seequence alignment revealed that it lacks the highly conserved catalytic residues of typical kinases, it has long been considered as a "pseudokinase" that lacks kinase activity.

*Carnitine O octyltransferase* (*CROT*) is located on the sheep chromosome 4 (34416632~34467060 bp). It is a peroxidase that converts short-chain fatty acids into carnitine in the mitochondrial matrix. In combination with carnitine palmityl transferase (*CPT1A*), fatty acid oxidation is performed. The *CROT* gene regulates fatty acid metabolism by

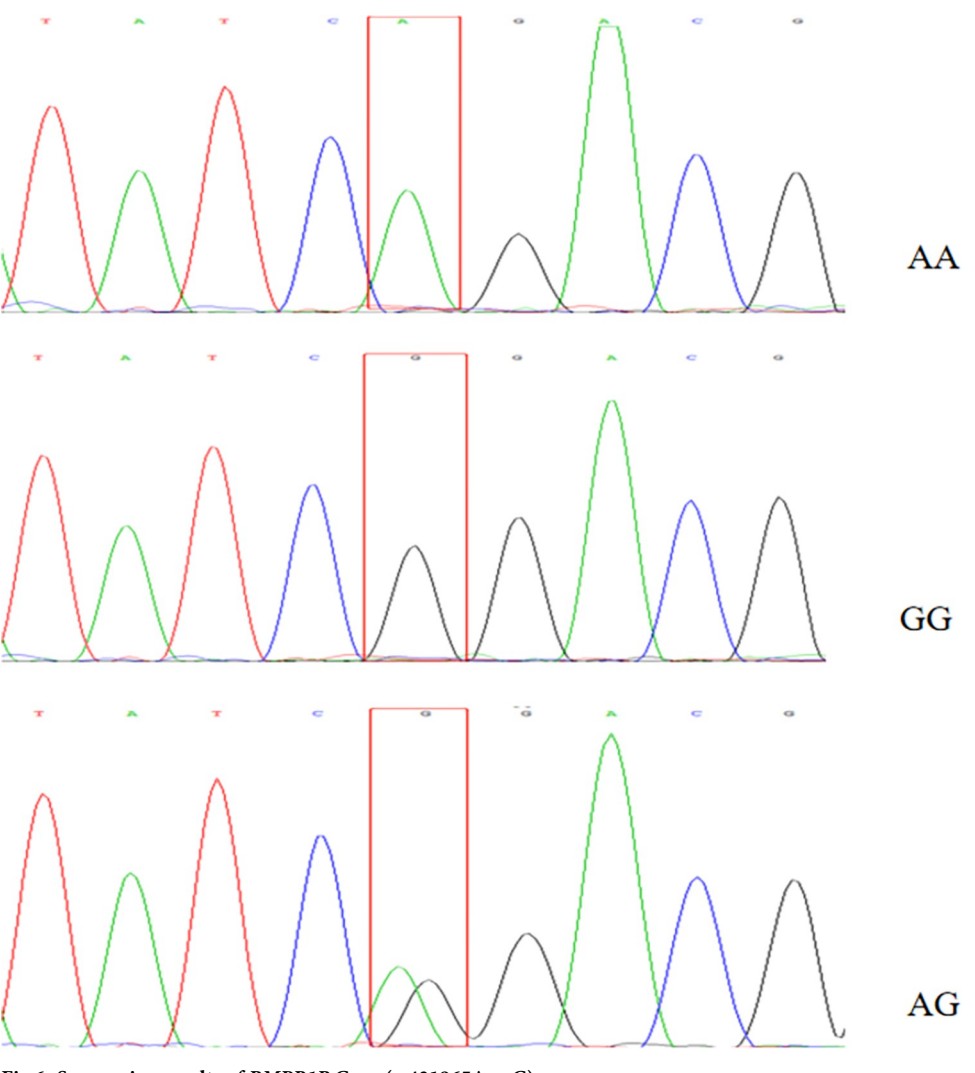

**Fig 6. Sequencing results of *BMPR1B* Gene (g.431965A > G).**

controlling the enzyme carnitine O-octyltransferase. The *CROT* gene in poultry, especially chickens, has been rarely reported. MiR-33 was found to target the silencing of fatty acid β oxidation-related genes *CROT* and *HADHB* [20,21].

Bone Morphogenetic protein receptor type 1B (*BMPR1B*) is located on the sheep chromosome 6 (30030664~30482585 bp). *BMPs* promote cartilage and bone tissue formation by binding to the bone morphogenetic protein receptor (*BMPR*) [22]. It also regulates biological functions, like cell proliferation, differentiation, migration and apoptosis, cell support, and embryonic development [23,24]. *BMPR1B*-induced follicular growth and development directly affects the egg production and is regulated by both autocrine and paracrine factors. The transforming growth factor β (*TGF-β*) family regulates follicular development through autocrine

**Table 4. Correlation analysis between *BMPRIB* Gene g.431965A > G locus and growth traits of Qira Black Sheep.**

| mutation | Traits | AA(55) | AG(28) | GG(6) |
|---|---|---|---|---|
| *BMPRIB*:g.431965ᵃ>G | Body height/kg | 44.06±7.75a | 39.25±8.45b | 38.03±5.06b |

action on the granulosa cells, oocytes, and follicular membrane cells [25–27]. Members of the family include bone morphogenetic proteins (*BMPs*), growth differentiation factors (GDFs), activins, and inhibitant. *BMPR1B* is a major gene affecting multiple lambs, which encodes the serine/threonine kinase activity containing receptor of multiple *BMPs*, which is vital in follicular development, growth, and immunity, and is closely related to animal reproduction [28]. In recent years, multiple studies have shown that the polymorphism of this gene is significantly associated with the early growth rate, body height, and body length of livestock [29].

*Fatty Acid Binding Protein 5* (*FABP5,e-FABP*), located on the sheep chromosome 9 (57575800~57580855 bp), is one of the important members of the fatty acid protein gene family. Knockdown of the *FABP5* gene leads to the cellular triglyceride accumulation, decreased cholesterol level, and decreased secretion of the ApoB100 protein and lipoprotein-like particles [30]. In studies of *FABP5* gene knockout mice, deletion of this gene caused sebaceous gland atrophy, sebum volume increase, and composition change [31]. Fang *et al*. [32] found in oral cancer cells that the overexpression of *FABP5* up-regulated the *MMP-9* gene expression, which increased cell proliferation and invasion. Estell *et al*. [33] found that this gene is closely related to the pig fat deposition, which was consistent with the results of Ojeda [34]. Therefore, *FABP5* gene is not only closely related to diverse cancer types, but also has been widely reported in the field of livestock and poultry molecular breeding research. However, no study has reported on the growth traits of sheep.

*Myeloid ectropic viral integration site 1* (*Meis1*) is located on the sheep chromosome 3 (41789220~41936042 bp). *Meis1* is an important regulator of cardiac differentiation and is involved in normal cardiac development. It also helps regulate the homeostasis of adult cardiac myocytes, and its mutation can also cause cardiac conduction defects [35]. In the process of cardiac differentiation, the expression patterns of *Meis1* and heart-specific gene *Nkx2.5* overlap, with the regulatory mechanisms of the two common targeted genes being consistent both temporally and spatially [36]. *Meis1* gene is not only involved in cell proliferation and differentiation, but also actively involved in blood vessel development and hematopoiesis, and its malfunctioning closely related to leukemia. *Meis1* is expressed in different solid tumors and is involved in the proliferation and differentiation of tumor cells. In angiogenesis and development, the *Meis1* gene is vital in the vascular endothelial growth factor (VEGF) pathway and hematopoietic stem cell (HSC) mediated angiogenesis.

*Ribosomal protein L17* (*RPL17*), is located on the sheep chromosome 4 (64829711~64830378 bp). It is a part of the large subunit, belonging to the ribosomal protein L22 family, which is the core member of the large subfamily of ribosomal protein. It is the only ribosomal protein that can interact with all six domains of the 23S rRNA, which is crucial for guiding the correct folding and conformation stability of 23S rRNA. It is also one of the six translocation binding sites on the surface of the ribosome. In addition, Paola *et al*. [37] found that *RPL17* was differentially expressed in the cleavage process of *Xenopus laevis*, and speculated that it might play a regulatory role in the cleavage. Zhong *et al*. [38] found that the *RPL17* gene expression level was different at different stages of the Japanese flounder embryogenesis, and it was the highest at the tail bud stage. The ribosomal protein gene was positively expressed during cell proliferation. In this study, *RPL17* was expressed in all tissues of *S. japonicus*, indicating that *RPL17* helps regulate the growth and development of tissue. Therefore, *RPL17* gene is highly expressed in the coelomic cells, and is important in regulating their development, cell division, proliferation, and differentiation.

*Ephrin A5* (*EFNA5*), a member of the receptor tyrosine kinases (*RTKs*) family, is located on the cell membrane. It is located on the sheep chromosome 5 (103981768~104274426 bp). Studies on the *EFNA5* gene in other fields are continuously increasing, with diverse new physiological functions already been reported. The mammalian *EFNA5* gene regulates proliferation,

apoptosis, differentiation, adhesion, and migration in cells [39]. It is involved in lens development and maintenance [40], osteosarcoma [41], and plays a role in follicular formation in female mammals [42]. Recent studies have found that the *EFNA5* gene can participate in fat thermogenesis and also regulate lipid metabolism [42], which may be a future therapeutic target for obesity and is also related to the meat quality traits of economic animals [43]. An *et al.* [43] conducted Genome-wide analysis of the six growth traits, including height, body length, hip height, heart size, abdominal size, and tube bone size in the Simmental cattle at three growth stages (6, 12 and 18 months after birth). They found multiple candidate genes, with *EFNA5* being one of them. Additionally, the *EFNA5* gene was also identified as a candidate gene affecting the growth traits of broilers [44]. These studies suggest that *EFNA5* may help regulate the meat quality traits of animals by promoting adipocyte differentiation. *EFNA5* also plays an important role in the individual development and physiological function [45].

*Nel-like type 1* (*NELL1*) is located on sheep chromosome 21 (20804714~21839088 bp), and is a novel growth factor associated with craniosynostosis with high specificity for bone and chondrocyte lines [46]. The *NELL1* gene was first isolated from the human fetal brain cDNA library, and it is important for promoting osteogenic differentiation and bone regeneration during the process of bone tissue development [47]. Recent studies have shown that [48–51] *NELL1* can promote the growth of osteogenic tissue, while osteogenesis and lipid formation are two opposite processes. Overexpression of *NELL1* promotes osteoblast differentiation and mineralization, while down regulation of *NELL1* inhibits osteoblast differentiation [52]. Multiple animal model studies have confirmed that *NELL1* can also promote bone and cartilage regeneration [53–55].

*Peroxisome proliferators-activated receptors-γ coactivator lA* (*PPARGC1A*), located on the sheep chromosome 6 (position 43944708~44662482 bp), is an important gene, which was first discovered and reported in the screening of the mouse brown fat cDNA library [56]. As a key nuclear transcription co-activator, *PPARGC1A* can bind to multiple transcription factors and participate in a series of orderly metabolic processes, thus being important in regulating mitochondrial biosynthesis, glucose metabolism, fatty acid oxidation, muscle fiber type transformation, etc [57–59]. *PPARGC1A* co-localized with the quantitative trait loci (*QTL*) associated with fatty acid synthesis and carcass traits [60,61]. Studies have shown that the *PPARGC1A* gene has SNPs that are significantly associated with the backfat thickness and meat color of suet [62–64]. The polymorphism loci of the *PPARGC1A* gene that were significantly associated with the growth traits (body weight, daily gain, etc.), had previously been detected in cattle, sheep, and chickens [65–67]. However, the correlation between the *PPARGC1A* gene and daily gain and feed conversion rate in pigs was rarely reported.

## Conclusion

In this study, we conducted an association analysis based on the SNPs data and growth traits of the Qira Black sheep and German Merino sheep. We obtained 55 significant SNPs loci, and there were a total of 84 genes near these SNPs. Mutant loci existed in the *BMPR1B* gene, and association analysis of growth traits with SNPs loci by SPSS 26.0 software revealed three genotypes at the g.431965A>G locus of the *BMPR1B* gene. Population validation of *BMPR1B* confirmed our study results. Therefore, it has both theoretical and practical significance for the molecular breeding of sheep growth performance.

## Supporting information

**S1 Table. Data for the top 1% of SNPs from small to large were obtained from GWAS of body length.**
(XLSX)

**S2 Table. Data for the top 1% of SNPs from small to large were obtained from GWAS of body height.**
(XLSX)

**S3 Table. Data for the top 1% of SNPs from small to large were obtained from GWAS of body Weight.**
(XLSX)

**S4 Table. Data for the top 1% of SNPs from small to large were obtained from GWAS of chest circumference.**
(XLSX)

# Acknowledgments

The authors would like to express their gratitude to EditSprings (https://www.editsprings.cn) for the expert linguistic services provided.

# Author Contributions

**Conceptualization:** Mirenisa Tuersuntuoheti, Shudong Liu.

**Data curation:** Mirenisa Tuersuntuoheti, Jihu Zhang, Cheng-long Zhang, Shudong Liu.

**Formal analysis:** Mirenisa Tuersuntuoheti, Cheng-long Zhang.

**Investigation:** Chunjie Liu, Qianqian Chang.

**Methodology:** Wen Zhou, Cheng-long Zhang.

**Resources:** Shudong Liu.

**Supervision:** Jihu Zhang, Shudong Liu.

**Writing – original draft:** Mirenisa Tuersuntuoheti.

**Writing – review & editing:** Mirenisa Tuersuntuoheti, Jihu Zhang, Shudong Liu.

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
