## [Decision Letter · Decision Letter 0]

17 Oct 2022

PONE-D-22-20448Searching for growth trait molecular markers in two sheep populations based on genome-wide association analysisPLOS ONE

Dear Dr. Liu,

Thank you for submitting your manuscript to PLOS ONE. After careful consideration, we feel that it has merit but does not fully meet PLOS ONE’s publication criteria as it currently stands. Therefore, we invite you to submit a revised version of the manuscript that addresses the points raised during the review process.

The three independent reviewers were very critics about your work and one proposed to reject the manuscript. Please take into consideration all comments (including those of reviewer #3) before resubmitting the revised version

We look forward to receiving your revised manuscript.

Kind regards,

Emidio Albertini, Ph.D.

Academic Editor

PLOS ONE

Journal Requirements:

2. Please amend either the abstract on the online submission form (via Edit Submission) or the abstract in the manuscript so that they are identical.

“This study was financially supported by grants from the National Natural Science Foundation of China (NO:32060743), Bintuan Science and Technology Program (2022CB001-09) and Efficient mutton sheep breed selection program ( xjnqry-g-2006 )”.

Upon re-submitting your revised manuscript, please upload your study’s minimal underlying data set as either Supporting Information files or to a stable, public repository and include the relevant URLs, DOIs, or accession numbers within your revised cover letter. For a list of acceptable repositories, please see http://journals.plos.org/plosone/s/data-availability#loc-recommended-repositories  Any potentially identifying patient information must be fully anonymized.

5. Please include a copy of Table 6 which you refer to in your text on page 7.

Reviewers' comments:

Reviewer's Responses to Questions

**Comments to the Author**

1. Is the manuscript technically sound, and do the data support the conclusions?

Reviewer #1: Yes

Reviewer #2: Yes

Reviewer #3: No

2. Has the statistical analysis been performed appropriately and rigorously? 

Reviewer #1: Yes

Reviewer #2: Yes

Reviewer #3: No

3. Have the authors made all data underlying the findings in their manuscript fully available?

Reviewer #1: Yes

Reviewer #2: No

Reviewer #3: Yes

4. Is the manuscript presented in an intelligible fashion and written in standard English?

Reviewer #1: Yes

Reviewer #2: Yes

Reviewer #3: No

5. Review Comments to the Author

Reviewer #1: In the present manuscript titled "Searching for growth trait molecular markers in two sheep populations based on genome-wide association analysis" by Mirenisa et al., 100 Qira black sheep and 84 German Merino sheep were randomly selected for DNA extraction via jugular vein to prepare Illumine Ovine SNP 50K Bead Chip. Genome-wide association studies (GWAS) were performed on sheep body weight, body height, body length and chest circumference using mixed linear models. 55 SNPs with significant correlation were obtained which were annotated by sheep reference genome and a total of 84 genes were obtained. BMPR1B was selected for population validation, and a correlation between FecB locus and body weight traits was found. This study provides supplementary work for molecular markers of sheep growth traits and has important theoretical significance and reference value for the mining of functional genes of sheep growth traits. This research is well-designed and interesting; however, there are still some concerns, which need to be improved by the authors.

1. Please check the grammer seriously, and the whole manuscript has to be proofread by a professional grammer agency.

2. Please check the references and unify the format of references.

3. The introduction part is too long, I suggested the writer should simplify this part. This part just listed some references.

4. “A total of 461,528 SNPs were obtained by genotyping.” You only used the SNP 50 chip, you only get 50K SNPs from each individual. The expression of this sentence is not clear.

5. How to explain the data deviation of the QQ plot? You should discuss this question in the discussion part. Your explanation of the QQ plot is not accepted. Please consider carefully the cause of the deviation of the QQ plot.

6. The number of sample individual is 184. As we know, the GWAS requires large sample size to ensure the accuracy of result.

7. “phenol-chloroform method” You need to provide more detailed steps or reference.

8. The GWAS analysis always used enrichment analysis of candidate genes and pathways using GO and KEGG. According to the SNPs how to find these candidate genes? You should provide more details.

9. “The genetic matrix was calculated by KING software and mapped by R language.” You should provide the name of the R package you used.

10. The abstract part needs to be modified. This part should not to specifically mention the analysis software you used.

11. In discussion part, you should discuss how to use these candidate genes for molecular markers in sheep genetics and breeding.

Reviewer #2: In the present manuscript titled "Searching for growth trait molecular markers in two sheep populations based on genome-wide association analysis" by Mirenisa et al., 100 Qira black sheep and 84 German Merino sheep were randomly selected for DNA extraction via jugular vein to prepare Illumine Ovine SNP 50K Bead Chip. Genome-wide association studies (GWAS) were performed on sheep body weight, body height, body length and chest circumference using mixed linear models. 55 SNPs with significant correlation were obtained which were annotated by sheep reference genome and a total of 84 genes were obtained. BMPR1B was selected for population validation, and a correlation between FecB locus and body weight traits was found. This study provides supplementary work for molecular markers of sheep growth traits and has important theoretical significance and reference value for the mining of functional genes of sheep growth traits. This research is well-designed and interesting; however, there are still some concerns, which need to be improved by the authors.

1. Please check the grammer seriously, and the whole manuscript has to be proofread by a professional grammer agency.

2. Please check the references and unify the format of references.

3. The introduction part is too long, I suggested the writer should simplify this part. This part just listed some references.

4. “A total of 461,528 SNPs were obtained by genotyping.” You only used the SNP 50 chip, you only get 50K SNPs from each individual. The expression of this sentence is not clear.

5. How to explain the data deviation of the QQ plot? You should discuss this question in the discussion part. Your explanation of the QQ plot is not accepted. Please consider carefully the cause of the deviation of the QQ plot.

6. The number of sample individual is 184. As we know, the GWAS requires large sample size to ensure the accuracy of result.

7. “phenol-chloroform method” You need to provide more detailed steps or reference.

8. The GWAS analysis always used enrichment analysis of candidate genes and pathways using GO and KEGG. According to the SNPs how to find these candidate genes? You should provide more details.

9. “The genetic matrix was calculated by KING software and mapped by R language.” You should provide the name of the R package you used.

10. The abstract part needs to be modified. This part should not to specifically mention the analysis software you used.

11. In discussion part, you should discuss how to use these candidate genes for molecular markers in sheep genetics and breeding.

Reviewer #3: The authors presented a manuscript on growth trait molecular markers in two sheep populations based on genome-wide association analysis. The topic of the investigation is interesting but there are several points of concern that make the paper not acceptable for the publication.

General comments:

-why the authors carried out their study using Qira black and German Merino breeds? Is there any etnological reason? This is a main problem. Without furnish a scientific reason about it, the the work seems unjustified, making the manuscript evaluation quite difficult.

-in the material and methods section several information are missing. Please remeber that this section has to furnish all the information to make the investigation repeatable.

-I am not able to understand why the authors genotyped the animals for polymorphisms at the BMPRIB gene… This is not specified in the aim of the work.

Specific comments:

-please if possibile include the line numbers that make the reviewer’s work easier.

-keywords: it is better to replace character with trait

-in the introduction section several sentences need references. Moreover the introduction section has to be shorted, avoiding information about cattle and focusing your attention on sheep. The list of the available SNPs chip in different specie is not updated.

Material and methods

-please supply information about the Qira black sheep breed, i.e. present census, number of farms, geographic distribution, productive purpose, under conservation plan? Were the animals sampled only females? In the statistical model is reported also “gender” and so I am confused about it..

-the authors reported “84 unrelated sheep” and few lines later they stated “no genealogical information”. The two statements are in opposition….. Please clarify. Same for “standardized feeding”: this is not adequate for a scientific paper. Please specify the feeding supplied to the animals.

-please specify at what age the morphological traits were measured.

-I am not able to understand what the authors mean with “experimental animals for population validation were 89 Qira black….”.

-please supply a reference for DNA extraction protocol.

Results

The authors genotyped all the animals using the Illumina Ovine 50K. In the first line of results they stated “a total of 461,528 SNPs were obtained..”… How is possible?

The main part of results is not about GWAS but about chromosome wide. This aspect has to be appropriately highlighted.

A true discussion section is missing. The main part of the discussion is a bibliographic review on gene onthology. The authors have to compare in this section the results of their study with the available literature on the same topic.

6. PLOS authors have the option to publish the peer review history of their article (what does this mean?). If published, this will include your full peer review and any attached files.

Reviewer #1: No

Reviewer #2: No

Reviewer #3: No

---

## [Author Response · Author response to Decision Letter 0]

18 Nov 2022

Journal Requirements:

　　Answer:We have modified the format of our paper according to the information on the website. Thank you very much for your reminder. 

2.Please amend either the abstract on the online submission form (via Edit Submission) or the abstract in the manuscript so that they are identical.

　Answer:We did have some inconsistencies in the submission of our papers. I will revise the paper submitted this time.

3. Thank you for stating the following financial disclosure:“This study was financially supported by grants from the National Natural Science Foundation of China (NO:32060743), Bintuan Science and Technology Program (2022CB001-09) and Efficient mutton sheep breed selection program ( xjnqry-g-2023 )”.

　Answer:This study was financially supported by grants from the National Natural Science Foundation of China (NO:32060743),Bintuan Science and Technology Program (2022CB001-09) and Efficient mutton sheep breed selection program ( xjnqry-g-2023 ). The authors thank Xinjiang agricultural area mutton sheep system for providing us with sheep genetic resources population and background information.

4.In your Data Availability statement, you have not specified where the minimal data set underlying the results described in your manuscript can be found. PLOS defines a study's minimal data set as the underlying data used to reach the conclusions drawn in the manuscript and any additional data required to replicate the reported study findings in their entirety. All PLOS journals require that the minimal data set be made fully available. For more information about our data policy, please see http://journals.plos.org/plosone/s/data-availability.

Upon re-submitting your revised manuscript, please upload your study’s minimal underlying data set as either Supporting Information files or to a stable, public repository and include the relevant URLs, DOIs, or accession numbers within your revised cover letter. For a list of acceptable repositories, please see http://journals.plos.org/plosone/s/data-availability#loc-recommended-repositories  Any potentially identifying patient information must be fully anonymized.

Answer:We have uploaded the original data after analysis to the attachment. There are four attachments in total.

5. Please include a copy of Table 6 which you refer to in your text on page 7.

Answer: According to the submission has been modified in the full text.

Reviewers' comments:

Reviewer's Responses to Questions

Comments to the Author

1. Is the manuscript technically sound, and do the data support the conclusions?

Reviewer #1: Yes

Reviewer #2: Yes

Reviewer #3: No

　

2. Has the statistical analysis been performed appropriately and rigorously?

Reviewer #1: Yes

Reviewer #2: Yes

Reviewer #3: No

3. Have the authors made all data underlying the findings in their manuscript fully available?

Reviewer #1: Yes

Reviewer #2: No

Reviewer #3: Yes

4. Is the manuscript presented in an intelligible fashion and written in standard English?

Reviewer #1: Yes

Reviewer #2: Yes

Reviewer #3: No

Review Comments to the Author

Answer:(1) We selected the growth trait phenotypes of two sheep breeds for genome-wide analysis because the genetic material for growth and development is the same. We conducted a verification experiment on the results of the later period. The result of verification is consistent with that of our GWAS. (2) We analyzed the data by constructing a mixed linear model. Verification is carried out according to statistical requirements.(3) After careful thinking, all the authors sent the original data to the submission system in the form of attached table, and finally appeared in the paper.(4)We have carefully revised the full text, and have found professional embellishments to revise the language.

1.Reviewer #1: In the present manuscript titled "Searching for growth trait molecular markers in two sheep populations based on genome-wide association analysis" by Mirenisa et al., 100 Qira black sheep and 84 German Merino sheep were randomly selected for DNA extraction via jugular vein to prepare Illumine Ovine SNP 50K Bead Chip. Genome-wide association studies (GWAS) were performed on sheep body weight, body height, body length and chest circumference using mixed linear models. 55 SNPs with significant correlation were obtained which were annotated by sheep reference genome and a total of 84 genes were obtained. BMPR1B was selected for population validation, and a correlation between FecB locus and body weight traits was found. This study provides supplementary work for molecular markers of sheep growth traits and has important theoretical significance and reference value for the mining of functional genes of sheep growth traits. This research is well-designed and interesting; however, there are still some concerns, which need to be improved by the authors.

Please check the grammer seriously, and the whole manuscript has to be proofread by a professional grammer agency.

Answer:The paper was polished by a professional institution. The supporting material has been placed in the corresponding document.

2. Please check the references and unify the format of references.

Answer:References have been modified in accordance with the document format of Plos one.

3. The introduction part is too long, I suggested the writer should simplify this part. This part just listed some references.

Answer:Some parts of the introduction have been removed. The deleted part is:In the GWAS study related to animal growth traits, An, et al. [2] detected 6 SNPs related to body height and length of Japanese wagyu cattle using Illumina CATTLE HD 770K SNP typing chip. These SNPs were located in or near 11 genes, five of which were novel candidate genes associated with body weight. An, et al. [3] used Illumina cattle 770K chip, combined with long-GWAS, single-trait GWAS and multi-trait GWAS, to study the changes of heart size, abdominal circumference, height, body length and tube circumference of Simmental beef cattle at different growth stages. The results showed that 58 significant SNPs were detected by the three models. 21 genes related to body size of Chinese cattle were matched. Through GWAS of backfat thickness in 304 Italian large white pigs, Fontanesi, et al.[4] found that PDE1C, CRISP1, STAT4 and STAB1 genes were significantly correlated with backfat thickness. Becker, et al. [5] selected 193 boars from the Swiss large white pig population after artificial insemination, and found one QTL locus on chromosome 14 and chromosome 2 for PH1 and carcass length, respectively, and two QTLS on chromosome 10 and chromosome 16 for hindleg length. Martin, et al. [6] conducted GWAS analysis of multiple nipples in 1185 alpine dairy goats and 810 Saneng dairy goats from France, and found that there were 17 SNPs with significant chromosome level on chromosome 10, but no SNPs with significant genomic level, which also indicated that multiple nipples were inherited in the form of multiple genes.

4. “A total of 461,528 SNPs were obtained by genotyping.” You only used the SNP 50 chip, you only get 50K SNPs from each individual. The expression of this sentence is not clear.

Answer: It should be 46,871 SNPs. This is my fault in work. I apologize to my esteemed editor.

5. How to explain the data deviation of the QQ plot? You should discuss this question in the discussion part. Your explanation of the QQ plot is not accepted. Please consider carefully the cause of the deviation of the QQ plot.

Answer:If there is deviation, it means that the actual value is deviated from the predicted value. In the GWAS study, if the SNP has a large deviation, it is believed that the deviation of the observed value of this SNP site is caused by the genetic effect of this SNP mutation.

6. The number of sample individual is 184. As we know, the GWAS requires large sample size to ensure the accuracy of result.

Answer:The data we collected from 184 sheep proved to be reliable. All of them were collected personally.

7. “phenol-chloroform method” You need to provide more detailed steps or reference.

Answer:Main steps of DNA extraction by phenol-chloroform method:

1. The animal tissue was placed in a 1.5-ml centrifuge tube and dealcoholized with 75%, 50% alcohol and pure water gradients, respectively. Each of the ladder Degree dehydration time is 5-10 min.

2. Place the tissue in a mortar, add appropriate amount, and rinse the grinding rod.

3. Add the ground tissue fluid to the seal of the heart tube with a pipette gun and put it into the shaker (56℃, 5h).

4. Add an equal volume of Tris saturated phenol (500 μl) and shake well (10 min).

5. Centrifugation: 12,000 rpm, 7 min, 4℃. After centrifugation, it is divided into three layers, the upper layer is DNA, the middle layer is protein, and the lower layer is organic matter.

6. Drain the top liquid and add it to the new centrifuge tube.

7. Tris saturated phenol: chloroform: isoamyl alcohol =25:24:1.

8. Add 450 μl of Tris saturated phenol, chloroform and isoamyl alcohol mixture to centrifuge tube containing supernatant and shake well for 10 min.

9. Centrifugation: 12,000 rpm, 7 min, 4℃.

10. Drain the supernatant and add it to a new centrifuge tube. Add an equal volume of 400 μl mixture of chloroform and isoamyl alcohol (chloroform: isoamyl alcohol = 24:1).

11. Centrifugation: 12,000 rpm, 7 min, 4℃.

12. Absorb the supernatant and add it to the new centrifuge tube. Add 2.5 times of 100% alcohol frozen at -20℃, overnight at -20℃.

13. The sample is taken out and centrifuged at 12,000 rpm, 7 min, 4℃.

14. Discard supernatant, leave white precipitate (DNA), add 75% alcohol frozen at -20℃, 400 μl, blow and dissolve repeatedly the solution.

15. Repeat step 14, 2 times (wash three times with 75% alcohol).

16. DNA extraction is complete.

We have cited reference to DNA extraction [1].

1. Sambrook J, Fritschi EF,Maniatis T.Molecular Cloning: A Laboratory Manual.Cold Spring Harbor Laboratory Press.1989;

8. The GWAS analysis always used enrichment analysis of candidate genes and pathways using GO and KEGG. According to the SNPs how to find these candidate genes? You should provide more details.

Answer:We used TASSEL5.0 software to analyze the SNP chip, and the SNP coordinate data came out for gene annotation. With the Sheep Genome Library (oar_rambouillet_v4.0 ftp://ftp.ncbi.nlm.nih.gov/genomes/all/GCF/002/742/125/GCF_002742125.1_oar_rambouillet_v4.0_genomic.fna.g), we found candidate genes.

9. “The genetic matrix was calculated by KING software and mapped by R language.” You should provide the name of the R package you used.

Answer:The name of the R package is qqMAN.

10. The abstract part needs to be modified. This part should not to specifically mention the analysis software you used.

Answer: We have modified the abstract and deleted the analysis software.

11.In discussion part, you should discuss how to use these candidate genes for molecular markers in sheep genetics and breeding.

Answer:

1. Find relevant marker genes according to breeding objectives.

2. Identify the marker genes and detect the polymorphic loci of the marker genes in the population.

3. The influence of marker genes on marker traits was analyzed.

4. Favorable genotypes were selected according to the analysis results.

2.Reviewer #2: In the present manuscript titled "Searching for growth trait molecular markers in two sheep populations based on genome-wide association analysis" by Mirenisa et al., 100 Qira black sheep and 84 German Merino sheep were randomly selected for DNA extraction via jugular vein to prepare Illumine Ovine SNP 50K Bead Chip. Genome-wide association studies (GWAS) were performed on sheep body weight, body height, body length and chest circumference using mixed linear models. 55 SNPs with significant correlation were obtained which were annotated by sheep reference genome and a total of 84 genes were obtained. BMPR1B was selected for population validation, and a correlation between FecB locus and body weight traits was found. This study provides supplementary work for molecular markers of sheep growth traits and has important theoretical significance and reference value for the mining of functional genes of sheep growth traits. This research is well-designed and interesting; however, there are still some concerns, which need to be improved by the authors.

Please check the grammer seriously, and the whole manuscript has to be proofread by a professional grammer agency.

Answer:The paper was polished by a professional institution. The supporting material has been placed in the corresponding document.

2. Please check the references and unify the format of references.

Answer:References have been modified in accordance with the document format of Plos one.

3. The introduction part is too long, I suggested the writer should simplify this part. This part just listed some references.

Answer:Some parts of the introduction have been removed. The deleted part is:In the GWAS study related to animal growth traits, An, et al. [2] detected 6 SNPs related to body height and length of Japanese wagyu cattle using Illumina CATTLE HD 770K SNP typing chip. These SNPs were located in or near 11 genes, five of which were novel candidate genes associated with body weight. An, et al. [3] used Illumina cattle 770K chip, combined with long-GWAS, single-trait GWAS and multi-trait GWAS, to study the changes of heart size, abdominal circumference, height, body length and tube circumference of Simmental beef cattle at different growth stages. The results showed that 58 significant SNPs were detected by the three models. 21 genes related to body size of Chinese cattle were matched. Through GWAS of backfat thickness in 304 Italian large white pigs, Fontanesi, et al.[4] found that PDE1C, CRISP1, STAT4 and STAB1 genes were significantly correlated with backfat thickness. Becker, et al. [5] selected 193 boars from the Swiss large white pig population after artificial insemination, and found one QTL locus on chromosome 14 and chromosome 2 for PH1 and carcass length, respectively, and two QTLS on chromosome 10 and chromosome 16 for hindleg length. Martin, et al. [6] conducted GWAS analysis of multiple nipples in 1185 alpine dairy goats and 810 Saneng dairy goats from France, and found that there were 17 SNPs with significant chromosome level on chromosome 10, but no SNPs with significant genomic level, which also indicated that multiple nipples were inherited in the form of multiple genes.

4. “A total of 461,528 SNPs were obtained by genotyping.” You only used the SNP 50 chip, you only get 50K SNPs from each individual. The expression of this sentence is not clear.

Answer: It should be 46,871 SNPs. This is my fault in work. I apologize to my esteemed editor.

5. How to explain the data deviation of the QQ plot? You should discuss this question in the discussion part. Your explanation of the QQ plot is not accepted. Please consider carefully the cause of the deviation of the QQ plot.

Answer:If there is deviation, it means that the actual value is deviated from the predicted value. In the GWAS study, if the SNP has a large deviation, it is believed that the deviation of the observed value of this SNP site is caused by the genetic effect of this SNP mutation.

6. The number of sample individual is 184. As we know, the GWAS requires large sample size to ensure the accuracy of result.

Answer:The data we collected from 184 sheep proved to be reliable. All of them were collected personally.

7. “phenol-chloroform method” You need to provide more detailed steps or reference.

Answer:Main steps of DNA extraction by phenol-chloroform method:

1. The animal tissue was placed in a 1.5-ml centrifuge tube and dealcoholized with 75%, 50% alcohol and pure water gradients, respectively. Each of the ladder Degree dehydration time is 5-10 min.

2. Place the tissue in a mortar, add appropriate amount, and rinse the grinding rod.

3. Add the ground tissue fluid to the seal of the heart tube with a pipette gun and put it into the shaker (56℃, 5h).

4. Add an equal volume of Tris saturated phenol (500 μl) and shake well (10 min).

5. Centrifugation: 12,000 rpm, 7 min, 4℃. After centrifugation, it is divided into three layers, the upper layer is DNA, the middle layer is protein, and the lower layer is organic matter.

6. Drain the top liquid and add it to the new centrifuge tube.

7. Tris saturated phenol: chloroform: isoamyl alcohol =25:24:1.

8. Add 450 μl of Tris saturated phenol, chloroform and isoamyl alcohol mixture to centrifuge tube containing supernatant and shake well for 10 min.

9. Centrifugation: 12,000 rpm, 7 min, 4℃.

10. Drain the supernatant and add it to a new centrifuge tube. Add an equal volume of 400 μl mixture of chloroform and isoamyl alcohol (chloroform: isoamyl alcohol = 24:1).

11. Centrifugation: 12,000 rpm, 7 min, 4℃.

12. Absorb the supernatant and add it to the new centrifuge tube. Add 2.5 times of 100% alcohol frozen at -20℃, overnight at -20℃.

13. The sample is taken out and centrifuged at 12,000 rpm, 7 min, 4℃.

14. Discard supernatant, leave white precipitate (DNA), add 75% alcohol frozen at -20℃, 400 μl, blow and dissolve repeatedly the solution.

15. Repeat step 14, 2 times (wash three times with 75% alcohol).

16. DNA extraction is complete.

We have cited reference to DNA extraction [1].

1. Sambrook J, Fritschi EF,Maniatis T.Molecular Cloning: A Laboratory Manual.Cold Spring Harbor Laboratory Press.1989;

8. The GWAS analysis always used enrichment analysis of candidate genes and pathways using GO and KEGG. According to the SNPs how to find these candidate genes? You should provide more details.

Answer:We used TASSEL5.0 software to analyze the SNP chip, and the SNP coordinate data came out for gene annotation. With the Sheep Genome Library (oar_rambouillet_v4.0 ftp://ftp.ncbi.nlm.nih.gov/genomes/all/GCF/002/742/125/GCF_002742125.1_oar_rambouillet_v4.0_genomic.fna.g), we found candidate genes.

9. “The genetic matrix was calculated by KING software and mapped by R language.” You should provide the name of the R package you used.

Answer:The name of the R package is qqMAN.

10. The abstract part needs to be modified. This part should not to specifically mention the analysis software you used.

Answer: We have modified the abstract and deleted the analysis software.

12.In discussion part, you should discuss how to use these candidate genes for molecular markers in sheep genetics and breeding.

Answer:

1. Find relevant marker genes according to breeding objectives.

2. Identify the marker genes and detect the polymorphic loci of the marker genes in the population.

3. The influence of marker genes on marker traits was analyzed.

4. Favorable genotypes were selected according to the analysis results.

1.Reviewer #3: The authors presented a manuscript on growth trait molecular markers in two sheep populations based on genome-wide association analysis. The topic of the investigation is interesting but there are several points of concern that make the paper not acceptable for the publication.

General comments:

-why the authors carried out their study using Qira black and German Merino breeds? Is there any etnological reason? This is a main problem. Without furnish a scientific reason about it, the the work seems unjustified, making the manuscript evaluation quite difficult.

Answer:The genetic law of growth and development traits in different sheep populations are the same. Growth and development traits are controlled by micro-effect polygenes. The body weight, body height, body length and chest circumference in our study are part of growth and development traits; and it is also controlled by micro-effect polygenes. When the two populations are analyzed together, growth and development traits can be more accurately and widely found in terms of genetic diversity.

2.-in the material and methods section several information are missing. Please remeber that this section has to furnish all the information to make the investigation repeatable.

Answer:We added the detailed information after the modification, to make the investigation repeatable.

3.-I am not able to understand why the authors genotyped the animals for polymorphisms at the BMPRIB gene… This is not specified in the aim of the work.

Answer:FecB is a molecular marker of BMPRIB, which was found by genome-wide association analysis (P=1.29×10-7, in weight trait). The aim of genotyping the animals for polymorphisms was to further verify the accuracy of genome-wide association analysis. We selected the results of this study for verification, and the verification result showed that there was a mutation site in BMPR1B gene, A > G mutation, and there were three genotypes: AA, AG and GG, which were consistent with our results after population verification.

Specific comments:

1.-please if possibile include the line numbers that make the reviewer’s work easier.

Answer:All the contents of the paper have been numbered.

2.-keywords: it is better to replace character with trait

Answer:The key words of the paper have been revised according to the opinion of the editor.

3.-in the introduction section several sentences need references. Moreover the introduction section has to be shorted, avoiding information about cattle and focusing your attention on sheep. The list of the available SNPs chip in different specie is not updated.

Answer:Some information about cattle and quotations in the introduction section have been deleted. The introduction has been simplified.

Material and methods

1.-please supply information about the Qira black sheep breed, i.e. present census, number of farms, geographic distribution, productive purpose, under conservation plan? Were the animals sampled only females? In the statistical model is reported also “gender” and so I am confused about it.

Answer:Qira black sheep breed has a long history, about 120 years of history. After long-term careful cultivation by local herdsmen, it has formed a local excellent sheep breed with stable genetic performance, large body, robust and adapted to arid desert ecological environment. It is mainly distributed in the Nuer ranch of Qira County, Hotan region, and there are only about 4,000 sheep left, which is a local protected sheep breed. It is only distributed in Qira County, Hotan region, mainly concentrated in Nuer ranch and the surrounding towns. The sheep breed has delicious meat and is rich in mineral elements, especially selenium, and the local government has made a breeding conservation plan. The animals sampled were all female, which reduced the influence of male and female factors on the growth and development of individual animals. We have made some modifications to the model of the species.

2.-the authors reported “84 unrelated sheep” and few lines later they stated “no genealogical information”. The two statements are in opposition….. Please clarify. Same for “standardized feeding”: this is not adequate for a scientific paper. Please specify the feeding supplied to the animals.

Answer:Sheep breeding experts in the southern region of Xinjiang, China, have standardized the forage materials for sheep forage breeding. The aim of adding 84 German Merino sheep is to better deal with genetic diversity errors. In this paper, we checked the samples and made modifications.

3.-please specify at what age the morphological traits were measured.

Answer:A total of 84 German Merino sheep (from Kezhou Breeding Farm) and 100 Qira black sheep (from Qira County Tianjin Aoqun Livestock Breeding Sheep Farm) among 1,200 ewes were genotyped using the Illumina 50K SNP panel for 54,241 markers. The body height, body length, weight, and chest circumference of the experimental animals were measured using electronic scales, measuring rods, and tape measures.

4.-I am not able to understand what the authors mean with “experimental animals for population validation were 89 Qira black….”.

Answer:We obtained 55 significant SNPs loci, among which 11 loci reached an extremely significant level. Population validation of BMPR1B confirmed our study results.Because after GWAS analysis, we verified the candidate genes and collected 89 Qira black sheep for population verification.

5.-please supply a reference for DNA extraction protocol.

Answer:In the materials and methods, we have cited a reference [1] to DNA extraction.

1.Sambrook J, Fritschi EF,Maniatis T.Molecular Cloning: A Laboratory Manual.Cold Spring Harbor Laboratory Press.1989;

Results

1.The authors genotyped all the animals using the Illumina Ovine 50K. In the first line of results they stated “a total of 461,528 SNPs were obtained..”… How is possible?

Answer: It should be 46,871 SNPs. This is my fault in work. I apologize to my esteemed editor.

2.The main part of results is not about GWAS but about chromosome wide. This aspect has to be appropriately highlighted.

Answer: It has been adjusted according to the pattern and requirements.The results are presented in the form of tables and figures.

3.A true discussion section is missing. The main part of the discussion is a bibliographic review on gene onthology. The authors have to compare in this section the results of their study with the available literature on the same topic.

Answer:Discussions have been held as requested.The false positive phenomenon of GWAS analysis was mainly due to population stratification, which eventually gave rise to multiple SNPs loci associated with the concerned traits in the GWAS analysis results [1]. To reduce these false-positive results, population stratification and relatedness among individuals were fully considered. We used PCA and breed effect to solve the population stratification phenomenon. As can be seen from the QQ-plot, there was no population stratification phenomenon in the corrected population, and the GWAS analysis results based on the MLM were relatively reliable. Additionally, the TASSEL 5.0 software provides a kinship matrix, two correction a group the genetic relationship between the two individuals[Please check this as the meaning is not clear.], to improve the effectiveness of the GWAS analysis. In this study, we performed a GWAS on the Qira black sheep and German Merino sheep via genotyping data by using a medium-density chip containing 46,871 SNPs. We detected 55 SNPs that were significantly associated with growth traits. From the genomic screening information, we found over 10% of the genes to be located in the sheep chromosomes 1, 2, 6, and 12. Among them, 10 loci were located on chromosome 1 (OAR1_223367286.1, S06270.1, S33133.1, etc.). There were 6 SNPs loci on chromosome 2, 6, and 12 (OAR2_179242768.1, OAR6_111740006.1, OAR12_60622021.1, etc.) We obtained a total of 84 genes through gene annotation. Based on gene enrichment analysis, we can preliminarily infer that these loci were important molecular markers that affected the growth and meat production traits of sheep.

 Zhang et al. [2] had obtained 36 significant SNPs for seven traits, while we obtained 55 significant SNPs for 4 traits (body weight, body height, body length, and chest circumference), which enriched the SNPs related to sheep growth and development. Furthermore, Jiang et al. [3] obtained 5 SNPs related to body height and 4 SNPs related to chest circumference in Hu sheep, whereas Hawlader et al. [4] obtained 39 SNPS related to body height in Australian Merino sheep. However, we identified 9 and 13 SNPs correlated with body height and chest circumference in this study, respectively. Since we had selected different germplasm resources for this study, we obtained the results at different loci. Although some of the annotated genes have already been previously reported, we found genes related to growth and development traits, including POMK, MEIS1, CROT, BMPR1B, FABP5, SPATA17, etc. Based on the NCBI and Gene Cards database annotations, we found that among the meat quality traits, genes related to growth traits mainly included POMK, BMPR1B, FABP5, RPL17, EFNA5, NELL1 and PPARGC1A.

1.Lander E, Kruglyak L. Genetic dissection of complex traits: guidelines for interpreting and reporting linkage results. Nat Genet. 1995;11:241-247.http://doi.org/10.1038/ng1195-241 PMID: 7581446.

2.Zhang L, Liu J, Zhao F, Ren H, Xu L ,Lu J, et al. Genome-Wide Association Studies for Growth and Meat Production Traits in Sheep. PLoS ONE.2013;8: e66569.http://doi.org/10.1371/journal.pone.0066569 PMID: 23825544

3.Jiang J, Cao Y, Shan H, Wu J, Song X, Jiang Y. The GWAS Analysis of Body Size and Population Verification of Related SNPs in Hu Sheep. Front Genet. 202;12:642552. http://doi.org/10.3389/fgene.2021.642552 PMID: 34093644

4.Al-Mamun HA, Kwan P, Clark SA, Ferdosi MH, Tellam R, Gondro C. Genome-wide association study of body weight in Australian Merino sheep reveals an orthologous region on OAR6 to human and bovine genomic regions affecting height and weight. Genet Sel Evol. 2015;47(1):66.http://doi.org/10.1186/s12711-015-0142-4 PMID:26272623

6. PLOS authors have the option to publish the peer review history of their article (what does this mean?). If published, this will include your full peer review and any attached files.

Do you want your identity to be public for this peer review? For information about this choice, including consent withdrawal, please see our Privacy Policy.

Reviewer #1: No

Reviewer #2: No

Reviewer #3: No

Answer: NO，Respect the privacy of authors and reviewers.

---

## [Decision Letter · Decision Letter 1]

27 Jan 2023

PONE-D-22-20448R1Exploring the growth trait molecular markers in two sheep populations based on genome-wide association analysisPLOS ONE

Dear Dr. Liu,

Thank you for submitting your manuscript to PLOS ONE. After careful consideration, we feel that it has merit but does not fully meet PLOS ONE’s publication criteria as it currently stands. Therefore, we invite you to submit a revised version of the manuscript that addresses the points raised during the review process.

We look forward to receiving your revised manuscript.

Kind regards,

Emidio Albertini, Ph.D.

Academic Editor

PLOS ONE

Reviewers' comments:

Reviewer's Responses to Questions

**Comments to the Author**

1. If the authors have adequately addressed your comments raised in a previous round of review and you feel that this manuscript is now acceptable for publication, you may indicate that here to bypass the “Comments to the Author” section, enter your conflict of interest statement in the “Confidential to Editor” section, and submit your "Accept" recommendation.

Reviewer #2: All comments have been addressed

Reviewer #3: (No Response)

2. Is the manuscript technically sound, and do the data support the conclusions?

Reviewer #2: Yes

Reviewer #3: Yes

3. Has the statistical analysis been performed appropriately and rigorously? 

Reviewer #2: Yes

Reviewer #3: Yes

4. Have the authors made all data underlying the findings in their manuscript fully available?

Reviewer #2: Yes

Reviewer #3: Yes

5. Is the manuscript presented in an intelligible fashion and written in standard English?

Reviewer #2: Yes

Reviewer #3: No

6. Review Comments to the Author

Reviewer #2: This manuscript is technically reliable. The author reflects the supporting data in the manuscript, and the statistical analysis is correct and strict. The language is concise and correct. I have no other problems and agree to receive the current version.

Reviewer #3: First of all I want to thank the authors for their efforts in improving the manuscript that in the present revised version is more understandable for the readers. The new version open also room for new comments.

General comment: in revising the manuscript please take care also about the editing aspects. There are still several editing problems in the new version.

Moreover, several times the use of technical definitions/terms is not adequate and some sentences are confused or unclear.

Finally, there are some further comments the I can suggest:

-title: the two populations studied are populations or breeds? It is important to take in consideration this aspect. The term population is usually used when no herd book is available. On the contrary it is better to use the term breed.

-line 29: “We obtained” is not adequate. May be “we identified”. Please check.

-line 32: “via the jugular vein” is for the blood collection and not for the DNA extraction. Moreover, what means “to prepare the Illumine Ovine SNP 50K Bead Chip”? May be “to genotype by using the Illumina Ovine SNP 50K Bead Chip”.

-line 32: please replace “quality control criteria were” with “quality control criteria for statistical analysis were”

-line 36: please use italic style for “Ovis aries” and add “genome” soon after Ovis aries.

-line 37: “we obtained a total of 84 genes”... this sentence is not complete. 84 genes for what? Associated to productive traits? Please specify it.

-line 46 and following lines: again I am not able to understand why the authors are referring to researches on deseases (in human) when there is a lot of literature available on the use of GWAS in animal genetics. My suggestion is to delete this part that it is out of contest.

-line 58: please replace “the sheep 50 K” with “the ovine 50 K”.

-line 107, please specify model and company for biophotometer.

-line 120: please supply a reference for the KING software

-line 123, please replace “GWAS analysis of four body weight traits in 184 sheep was performed using Illumina OvineSNP50 microarray” with “GWAS analysis of four body weight traits in 184 sheep was performed using genotypes obtained from Illumina OvineSNP50 microarray”.

-line 137: please check the use of “P” with capital letter and without in the same line.

-line 139: please avoid paragraph so short. It could be better to merge this one with the previous one.

-line 144: please supply a reference for Primer Premier 6.0 software.

-line 148, please specify the country for Shanghai Shenggong Bioengineering Company.

-table 1: please replace the title of the first column with “Gene”.

-line 151: about the PCR conditions: please mention the final/stock concentrations of each reagent, without which it’s meaningless.

-line 158, please supply a reference for DNASTAR

-line 159: please supply a reference for SPSS 26.0 software

-line 160 please replace coma with dot.

-line 176: German-American sheep or German Merino sheep? Please clarify.

-line 184: there are typos in this line, please check.

-line 190: the sentence “with significant GWAS analysis results” is not correct. May be “with significant associations with productive traits obtained in the GWAS analysis”

-line 213, Qira with capital letter.

Table 3: please use the italic style for all the gene acronyms (check in all the manuscript long).

-line 223: what is “Cele black sheep”??

-line 227, please replace “amplified” with “amplified region of”

-line 230, gene acronym in italic style as in the following line.

-line 239: in my opinion this sentence is not complete: “As can be seen from Table 4, the BMPR1B gene g.431965A>G in the Qira black sheep”

-line 241: was the significance level the same among the different genotypes? In the text seems to be 0.05 but in the Table 4 there are two different levels (a and b). Please clarify.

-line 379: it could be better to replace “genome microchips” with “SNPs data”

Finally try to improve the conclusion section with a message to take at home for the breeders: try to explain the transferability of your results and in what way these results can be important in animal breeding of the studied breeds.

7. PLOS authors have the option to publish the peer review history of their article (what does this mean?). If published, this will include your full peer review and any attached files.

Reviewer #2: No

Reviewer #3: No

---

## [Author Response · Author response to Decision Letter 1]

22 Feb 2023

Reviewer #2:This manuscript is technically reliable.The author reflects the supporting data in the

manuscript, and the statistical analysis is correct and strict. The language is concise and correct.I have no other problems and agree to receive the current version.

Reviewer #3:First of all I want to thank the authors for their efforts in improving the manuscript that in the

present revised version is more understandable for the readers. The new version open also room for new comments.

General comment: in revising the manuscript please take care also about the editing aspects. There are still

several editing problems in the new version .

---

## [Decision Letter · Decision Letter 2]

8 Mar 2023

Exploring the growth trait molecular markers in two sheep breeds based on Genome-wide association analysis

PONE-D-22-20448R2

Dear Dr. Liu,

We’re pleased to inform you that your manuscript has been judged scientifically suitable for publication and will be formally accepted for publication once it meets all outstanding technical requirements.

Kind regards,

Emidio Albertini, Ph.D.

Academic Editor

PLOS ONE

Additional Editor Comments (optional):

Reviewers' comments:

Reviewer's Responses to Questions

**Comments to the Author**

1. If the authors have adequately addressed your comments raised in a previous round of review and you feel that this manuscript is now acceptable for publication, you may indicate that here to bypass the “Comments to the Author” section, enter your conflict of interest statement in the “Confidential to Editor” section, and submit your "Accept" recommendation.

Reviewer #2: All comments have been addressed

Reviewer #3: All comments have been addressed

2. Is the manuscript technically sound, and do the data support the conclusions?

Reviewer #2: Yes

Reviewer #3: Yes

3. Has the statistical analysis been performed appropriately and rigorously? 

Reviewer #2: Yes

Reviewer #3: Yes

4. Have the authors made all data underlying the findings in their manuscript fully available?

Reviewer #2: Yes

Reviewer #3: Yes

5. Is the manuscript presented in an intelligible fashion and written in standard English?

Reviewer #2: Yes

Reviewer #3: Yes

6. Review Comments to the Author

Reviewer #2: (No Response)

Reviewer #3: Thanks to the authors for their efforts in improving the manuscript that in the present revised version is now more understandable for the readers.

7. PLOS authors have the option to publish the peer review history of their article (what does this mean?). If published, this will include your full peer review and any attached files.

Reviewer #2: No

Reviewer #3: No

---

## [Editor Report · Acceptance letter]

14 Mar 2023

PONE-D-22-20448R2 

Exploring the growth trait molecular markers in two sheep breeds based on Genome-wide association analysis 

Dear Dr. Liu:

I'm pleased to inform you that your manuscript has been deemed suitable for publication in PLOS ONE. Congratulations! Your manuscript is now with our production department. 

Kind regards, 

on behalf of

Prof. Emidio Albertini 

Academic Editor

PLOS ONE